# THE PITFALLS AND PROMISE OF CONFORMAL INFERENCE UNDER ADVERSARIAL ATTACKS

## ABSTRACT

In safety-critical applications such as medical imaging and autonomous driving, where decisions have profound implications for patient health and road safety, it is imperative to maintain both high adversarial robustness to protect against potential adversarial attacks and reliable uncertainty quantification in decision-making. With extensive research focused on enhancing adversarial robustness through various forms of improved adversarial training (AT), a notable knowledge gap remains concerning the uncertainty inherent in adversarially trained models. To address this gap, this study investigates the uncertainty of deep learning models by examining the performance of conformal prediction (CP) in the context of standard adversarial attacks within the adversarial defense community. It is first unveiled that existing conformal prediction methods fail under the commonly used $l_\infty$-norm bounded attack if the model is not adversarially trained, which underpins the importance of adversarial training for CP. Our paper next demonstrates that the prediction set size of CP using adversarially trained models with AT variants is often worse than using standard AT, which inspires us to research into CP-efficient AT for improved prediction set size. Our empirical study finds two factors are strongly correlated with the efficiency of CP: 1) *predictive entropy* and 2) *distribution of the true class probability ranking (TCPR)*. Based on the two observations, we propose the Uncertainty-Reducing AT (AT-UR) to learn an adversarially robust and CP-efficient model with *entropy minimization* and *Beta importance weighting*. Theoretically, this paper presents generalization error analysis for Beta importance weighting indicating that the proposed UR-AT can potentially learn a model with improved generalization. Empirically, we demonstrate the substantially improved CP-efficiency of our method on four image classification datasets compared with several popular AT baselines.

## 1 INTRODUCTION

The research into adversarial defense has been focused on improving adversarial training with various strategies, such as logit-level supervision (Zhang et al., 2019; Cui et al., 2021a) and loss reweighting (Wang et al., 2019; Liu et al., 2021a). However, the predictive uncertainty of an adversarially trained model is a crucial dimension of the model in safety-critic applications such as healthcare (Razzak et al., 2018), and is not sufficiently understood. Existing works focus on calibration uncertainty (Stutz et al., 2020; Qin et al., 2021; Kireev et al., 2022), without investigating a practical uncertainty quantification of a model, e.g., a prediction set in image classification (Shafer & Vovk, 2008; Angelopoulos et al., 2020; Romano et al., 2020).

On the other hand, the research into conformal prediction (CP) has been extended to non-i.i.d. (identically independently distributed) settings, including distribution shifts (Gibbs & Candes, 2021) and toy adversarial noise (Ghosh et al., 2023; Gendler et al., 2021). However, there is little research work on the performance of CP under standard adversarial attacks in the adversarial defense community, such as PGD-based attacks (Madry et al., 2018; Croce & Hein, 2020) with $l_\infty$-norm bounded perturbations. For example, Gendler et al. (2021) and Ghosh et al. (2023) only consider $l_2$-norm bounded adversarial perturbations with a small attack budget, e.g., $\epsilon = 0.125$ for the CIFAR dataset (Krizhevsky et al., 2009). In contrast, the common $l_2$-norm bounded attack budget in adversarial defense community reaches $\epsilon = 0.5$ on CIFAR (Croce & Hein, 2020). In other words, existing research on adversarially robust conformal prediction is not practical enough to be used under standard adversarial attacks.

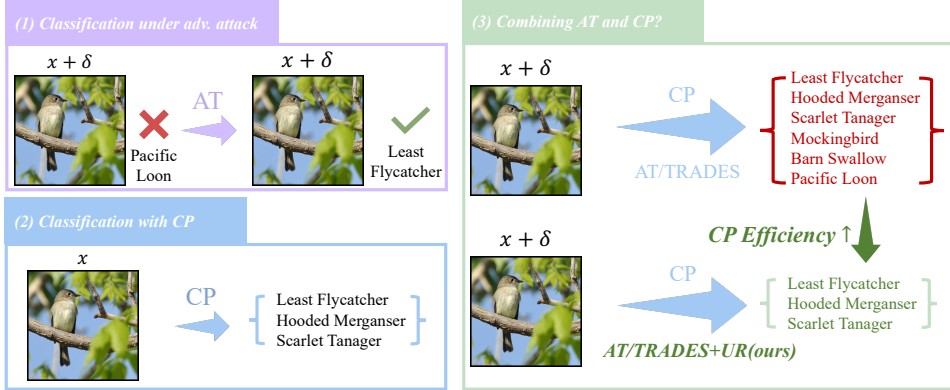

Figure 1: The proposed uncertainty-reducing adversarial training (AT-UR) improves the CP-efficiency of existing adversarial training methods like AT, FAT and TRADES. (1) AT improves the Top-1 robust accuracy of a standard model; (2) CP generates a prediction set with a pre-specified coverage guarantee for an input image, but for models not adversarially trained, CP fails to generate informative prediction sets, as the prediction set size is almost the same as the class number, when models being attacked (Fig. 2); (3) When using CP in an adversarially trained model, the prediction set size is generally large, leading to inefficient CP. Our AT-UR substantially improves the CP-efficiency of existing AT methods.

In this context, our paper is among the first research papers to explore uncertainty of deep learning models within the framework of CP in the presence of a *standard* adversary. We first present an empirical result that shows the failure of three popular CP methods on non-robust models under a standard adversarial attack, indicating the necessity of using adversarial training (AT) during the training stage. Next, we show the CP performance of three popular AT methods and find that advanced AT methods like TRADES (Zhang et al., 2019) and MART (Wang et al., 2019) substantially increase the prediction set size in CP even though they improve the Top-1 robust accuracy. This key observation inspires us to develop the uncertainty-reducing AT (AT-UR) to learn an adversarially robust model with improved *CP-efficiency* (Angelopoulos et al., 2020), meaning that CP uses a smaller prediction set size to satisfy the coverage. The proposed AT-UR consists of two training techniques based on our observation about the two major factors that affect prediction set sizes: prediction entropy and the True Class Probability Ranking (TCPR), both defined in Sec. 5. Our AT-UR is demonstrated to be effective at reducing the prediction set size of models on multiple image classification datasets. In summary, there are four major contributions of this paper.

1. We test several CP methods under commonly used adversarial attacks in the adversarial defense community. It turns out that but for models not adversarially trained, CP cannot to generate informative prediction sets. Thus, adversarial training is necessary for CP to work under adversarial attacks.

2. We test the performance of adversarially trained models with CP and demonstrate that improved AT often learns a more uncertain model and leads to less efficient CP with increased prediction set sizes.

3. Our empirical study reveals that the prediction set size of a model is closely related to the entropy of predictive distributions and the distribution of true class probability ranking (TCPR). Inspired by the empirical observation, we propose uncertainty-reducing AT (AT-UR) to learn a CP-efficient and adversarially robust model by minimizing the entropy of predictive distributions and weighting losses based on TCPR with a Beta density function.

4. Our theoretical study provides a generalization bound for the proposed Beta weighting training that shows this weighting scheme will potentially learn a model with improved generalization. Our empirical study demonstrates that the proposed AT-UR learns adversarially robust models with substantially improved CP-efficiency on four image classification datasets.

The paper has a structure as follows. Section 2 discusses related works and Section 3 introduces mathematical notations and two key concepts in this paper. Section 4 shows the pitfalls of three CP methods under standard attacks when the model is not robustly trained and the low CP-efficiency of two improved AT methods and motivates us to develop the AT-UR introduced in Section 5. Our major empirical results are shown in Section 6 and we conclude the paper in Section 7.

## 2 RELATED WORKS

**Adversarial Robustness.** The most effective approach to defending against adversarial attacks is adversarial training (AT) (Madry et al., 2018). There is a sequence of works following the vanilla

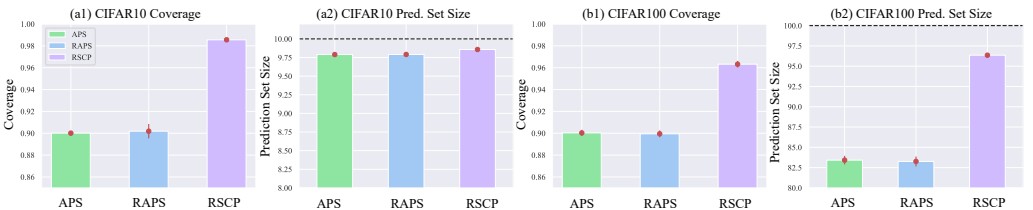

Figure 2: The performance of three representative CP methods using non-robust models under standard adversarial attacks in the adversarial defense community. The red line denotes means standard deviation of the metric. For comparison, the average prediction set sizes for normal images is 1.03 and 2.39 for CIFAR10 and CIFAR 100. See Sec. 6.1 for details of the experiment.

version of AT based on projected gradient descent (PGD), including regularization (Qin et al., 2019; Liu & Chan, 2022; Liu et al., 2021b), logit-level supervision (Zhang et al., 2019; Cui et al., 2021a) and loss re-weighting (Wang et al., 2019; Liu et al., 2021a). Existing methods on regularization focus on improving Top-1 robust accuracy by training the model with certain properties like linearization (Qin et al., 2019) and large margins (Liu & Chan, 2022). In contrast, our work focuses on the prediction set size, i.e., the efficiency of CP, in adversarially trained models by regularizing the model to have low prediction entropy. The entropy minimization regularization also entails logit-level supervision as in Zhang et al. (2019). In comparison, our proposed approach, AT-EM, enhances CP efficiency, whereas TRADES (Zhang et al., 2019) impedes CP-efficiency. The most related work is Gendler et al. (2021) which also studies CP under adversarial attacks. However, there are two fundamental differences: 1) Gendler et al. (2021) only considers a small attack budget under $l_2$-norm bounded attacks, while our work investigates CP under common adversarial attacks in adversarial defense literature with $l_\infty$-norm bounded attacks; 2) Our paper shows that AT is essential for CP to work under strong adversarial attacks and proposes novel AT methods to learn a CP-efficient and adversarially-robust model, while Gendler et al. (2021) only considers the post-training stage. Our experiment validates that Gendler et al. (2021) fails when there are strong adversarial attacks (Fig. 2).

**Uncertainty Quantification.** Uncertainty quantification aims to provide an uncertainty measure for a machine learning system's decisions. Within this domain, Bayesian methods stand out as a principled approach, treating model parameters as random variables with distinct probability distributions. This is exemplified in Bayesian Neural Networks (BNNs), which place priors on network weights and biases, updating these with posterior distributions as data is observed (Gal & Ghahramani, 2016; Kendall & Gal, 2017; Cui et al., 2020). However, the large scale of modern neural networks introduces challenges for Bayesian methods, making piror and posterior selection, and approximate inference daunting tasks (Kingma et al., 2015; Cui et al., 2021b; 2023; CUI et al., 2023). This can sometimes compromise the optimal uncertainty quantification in BNNs. In contrast, the frequentist approach offers a more direct route to uncertainty estimation. It views model parameters as fixed yet unknown, deriving uncertainty through methods like conformal prediction (Vovk et al., 1999; Ghosh et al., 2023; Gendler et al., 2021). While Bayesian methods integrate prior beliefs with data, their computational demands in large networks can be overwhelming, positioning the straightforward frequentist methods as a viable alternative for efficient uncertainty quantification. Thus, our paper investigates the uncertainty of adversarially trained models via CP. Note that our work is fundamentally different from existing research on uncertainty calibration for AT (Stutz et al., 2020; Qin et al., 2021; Kireev et al., 2022), as our focus is to produce a valid prediction set while uncertainty calibration aims to align accuracy and uncertainty. Finally, Einbinder et al. (2022) proposes to train a model with uniform conformity scores on a calibration set in standard training, while our work proposes CP-aware adversarial training to reduce prediction set sizes.

## 3 PRELIMINARY

Before diving into the details of our analysis and the proposed method, we first introduce our mathematical notations, adversarial training and conformal prediction.

**Notations.** Denote a training set with $m$ samples by $\mathcal{D}_{\text{tr}} = \{(x_i, y_i)\}_{i=1}^m$. Suppose each data sample $(x_i, y_i) \in \mathcal{X} \times \mathcal{Y}$ is drawn from an underlying distribution $\mathcal{P}$ defined on the space $\mathcal{X} \times \mathcal{Y}$, where $x_i$ and $y_i$ are the feature and label, respectively. Particularly, we consider the classification problem and assume that there are $K$ classes, i.e., $\mathcal{Y} = \{1, ..., K\}$ (we denote $[K] = \{1, ..., K\}$ for simplicity). Let $f_\theta : \mathcal{X} \to \Delta_p^K$ denote a predictive model from a hypothesis class $\mathcal{F}$ that generates a $K$-dimensional probability simplex: $\Delta_p^K = \{v \in [0, 1]^K : \sum_{k=1}^K v_k = 1\}$. $\theta$ is the model parameter we optimize during training. A loss function $\ell : \mathcal{Y} \times \mathcal{Y} \to \mathbb{R}$ is used to measure the difference between the prediction made by $f_\theta(x)$ and the ground-truth label $y$.

|  | Dataset | CIFAR10 | CIFAR100 | Caltech256 | CUB200 |
|---|---|---|---|---|---|
| AT | Rob. Coverage | 90.55(0.51) | 90.45(0.59) | 91.35(0.85) | 90.33(0.89) |
|  | Rob. Set Size | **3.10(0.07)** | **23.79(0.80)** | **43.20(2.11)** | **37.37(2.11)** |
|  | Clean Acc. | 89.76(0.15) | 68.92(0.38) | 75.28(0.51) | 65.36(0.27) |
|  | Rob. Acc. | 50.17(0.91) | 28.49(1.14) | 47.53(0.67) | 26.29(0.44) |
| TRADES | Rob. Coverage | 90.72(0.62) | 90.35(0.57) | 90.82(0.81) | 90.38(0.76) |
|  | Rob. Set Size | 3.31(0.09) | 27.60(0.97) | 44.80(3.42) | 52.18(2.60) |
|  | Clean Acc. | 87.31(0.27) | 62.83(0.33) | 69.57(0.25) | 58.16(0.38) |
|  | Rob. Acc. | 53.07(0.23) | 32.07(0.20) | 47.07(0.37) | 27.82(0.23) |
| MART | Rob. Coverage | 91.60(0.48) | 90.67(0.83) | 91.92(0.84) | 90.22(0.53) |
|  | Rob. Set Size | 3.81(0.07) | 28.37(1.29) | 46.79(2.73) | 45.31(1.78) |
|  | Clean Acc. | 85.43(0.24) | 59.66(0.26) | 69.68(0.31) | 58.72(0.18) |
|  | Rob. Acc. | **54.48(0.29)** | **34.04(0.46)** | **49.82(0.32)** | **28.99(0.25)** |

Table 1: CP and Top-1 accuracy of three popular adversarial defense methods under PGD100 adversarial attack. Bold numbers are the best prediction set size and robust accuracy.

To measure the performance of $f_\theta$ in the sense of population over $\mathcal{P}$, the *true risk* is typically defined as $R(f_\theta) = \mathbb{P}_{(x,y)\sim\mathcal{P}}[f_\theta(x) \neq y]$. Unfortunately, $R(f)$ cannot be realized in practice, since the underlying $\mathcal{P}$ is unreachable. Instead, the *empirical risk* $\widehat{R}(f_\theta) = \frac{1}{m}\sum_{i=1}^{m} \mathbb{I}[f_\theta(x_i) \neq y_i]$ is usually used to estimate $R(f_\theta)$, where $\mathbb{I}[\cdot]$ is the indicator function. The estimation error of $\widehat{R}(f_\theta)$ to $R(f_\theta)$ is usually referred to as generalization error bound and can be bounded by a standard rate $O(1/\sqrt{m})$. To enable the minimization of empirical risk, a loss function $\ell$ is used as the surrogate of $\mathbb{I}[\cdot]$, leading to the classical learning paradigm empirical risk minimization (ERM): $\min_{f_\theta \in \mathcal{F}} \widehat{L}(f_\theta) = \frac{1}{m}\sum_{i=1}^{m} \ell(f_\theta(x_i), y_i)$. In this work, we use the standard cross-entropy loss as the loss function,

$$\ell(f_\theta(x_i), y_i) = -\sum_{j=1}^{K} y_{ij} \log(f_\theta(x_i)_j). \tag{1}$$

**Adversarial training.** Write the loss for sample $(x_i, y_i)$ in adversarial training as $l(\tilde{x}_i, y_i)$, where $\tilde{x}_i = x_i + \delta_i$ and $\delta_i$ is generated from an adversarial attack, e.g., PGD attack (Madry et al., 2018). The vanilla adversarial training minimizes the loss with uniform weights for a mini-batch with $B$ samples, i.e.,

$$\nabla f_\theta = \nabla \frac{1}{B} \sum_{i=1}^{B} l(f_\theta(\tilde{x}_i)_j, y_i), \tag{2}$$

where $\nabla f_\theta$ is the gradient in this mini-batch step optimization with respect to $\theta$.

**Conformal prediction (CP).** CP is a distribution-free uncertainty quantification method and can be used in a wide range of tasks including both regression and classification (Vovk et al., 1999; 2005). This paper focuses on the image classification task, where CP outputs a prediction set instead of the Top-1 predicted class as in a standard image classification model, and satisfies a coverage guarantee. Mathematically, CP maps an input sample $x$ to a prediction set $\mathcal{C}(x)$, which is subset of $[K] = \{1, \cdots, K\}$, with the following coverage guarantee,

$$P(y \in \mathcal{C}(x)) \geq 1 - \alpha, \tag{3}$$

where $1 - \alpha$ is a pre-defined confidence level such as 90%, meaning that the prediction set will contain the ground-truth label with 90% confidence for future data. This paper mainly considers the *split conformal prediction*, an efficient CP approach applicable to any pre-trained black-box classifier (Papadopoulos et al., 2002; Lei et al., 2018) as it does not need to re-train the classifier with different train-calibration-test splits.

The prediction set of CP is produced by the calibration-then-test procedure. In the context of a classification task, we define a prediction set function $\mathcal{S}(x, u; \pi, \tau)$, where $u$ is a random variable sampled from a uniform distribution Uniform$[0, 1]$ independent of all other variables, $\pi$ is shorthand for the predictive distribution $f_\theta(x)$, and $\tau$ is a threshold parameter that controls the size of the prediction set. An increase in the value of $\tau$ leads to an expansion in the size of the prediction set within $\mathcal{S}(x, u; \pi, \tau)$. We give one example (Romano et al., 2020) of the function $\mathcal{S}$ in Appendix A. The calibration process computes the smallest threshold parameter $\hat{\tau}_{\text{cal}}$ to achieve an empirical coverage of $(1 - \alpha)(n_c + 1)/n_c$ on the calibrations set with $n_c$ samples. For a test sample $x^*$, a prediction set is the output of the function $\mathcal{S}(x^*, u; \pi^*, \hat{\tau}_{\text{cal}})$.

## 4 NECESSITATE AT FOR ROBUST AND EFFICIENT COVERAGE.

**The pitfalls of CP under strong adversarial attacks**. We test the performance of three conformal prediction methods, i.e., APS (Adaptive Prediction Sets) (Romano et al., 2020), RAPS (Regularized

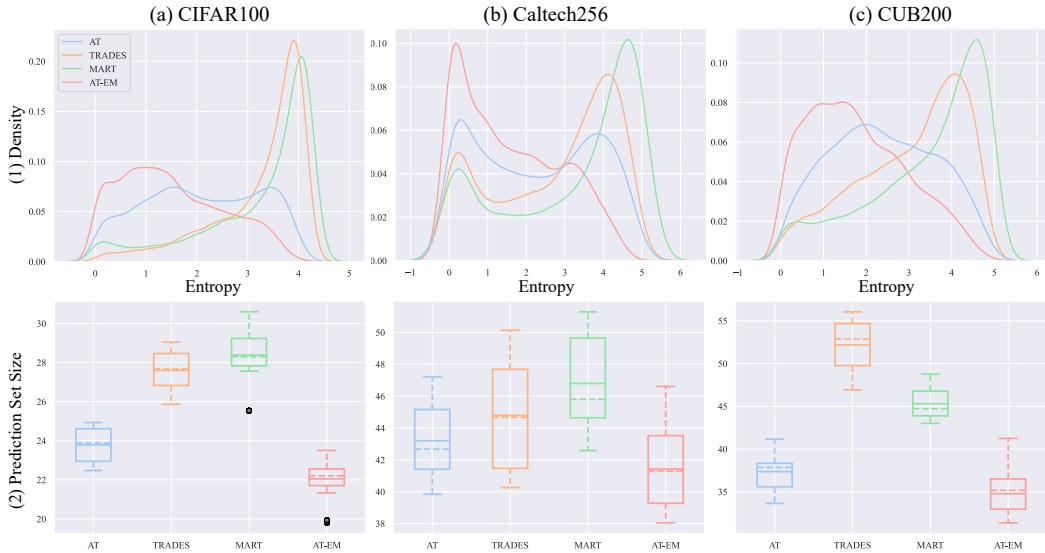

Figure 3: **(1)**: The kernel density estimation for predictive distribution's entropy on adversarial test sets. **(2)**: Box plot of prediction set size of three AT baselines and AT-EM. AT-EM effectively controls prediction entropy and improves CP-efficiency. See Tab. 1 and Tab. 2 for their coverages.

Adaptive Prediction Sets) (Angelopoulos et al., 2020) and RSCP (Randomly Smoothed Conformal Prediction) (Gendler et al., 2021), under standard adversarial attacks. Specifically, for APS and RAPS, we use PGD100 adversarial attacks with $l_\infty$-norm bound and attack budget $\epsilon = 8/255 = 0.0314$. For RSCP, we adopt PGD20 with an $l_2$-norm bound, in accordance with the original paper's settings, but with a larger attack budget of $\epsilon = 0.5$ as in RobustBench (Croce & Hein, 2020). If not specified otherwise, we use adversarial attack PGD100 with $l_\infty$ norm and $\epsilon = 8/255 = 0.0314$ to generate adversarial examples throughout this paper.

Fig. 2 shows the coverage and prediction set size of three CP methods on CIFAR10 and CIFAR100 when models are trained in a standard way, i.e., without adversarial training. Although all CP methods have good coverages, their prediction set sizes are close to the number of classes in both datasets as the classifier is completely broken under the strong adversarial attacks. In contrast, when the same models are applied to standard images, the prediction set sizes are 1.03 and 2.39 for CIFAR10/CIFAR100. This result reveals that adversarial training is indispensable if one wants to use CP to get reasonable uncertainty quantification for their model in an adversarial environment. Therefore, in next section, we test AT and two improved AT methods to investigate the performance of CP for adversarially trained models.

**Improved AT Compromises Conformal Prediction's Efficiency**. We test three popular adversarial training methods, i.e., AT (Madry et al., 2018), TRADES (Zhang et al., 2019) and MART (Wang et al., 2019), using APS as the conformal prediction method under a commonly used adversarial attack, PGD100 with $l_\infty$-norm and $\epsilon = 8/255 = 0.0314$. See more detailed experimental settings in Sec. 6. Tab. 1 shows their coverage and prediction set size, as well as clean and robust accuracy on four datasets. The results demonstrate that while the two enhanced adversarial training methods, TRADES and MART, effectively improve the Top-1 accuracy in the presence of adversarial attacks, they lead to an increase in the size of the prediction set, consequently yielding a less CP-efficient model. In other words, the improvement in Top-1 accuracy does not necessarily lead to less uncertainty. Therefore, to design a new AT method that learns an adversarially robust model with efficient CP, a deep investigation into the prediction set size is necessary. In the following section, we identify two major factors that play an important role in controlling the prediction set size through our empirical study.

## 5 UNCERTAINTY-REDUCING ADVERSARIAL TRAINING

This section investigates two factors highly correlated with prediction set size and introduces our uncertainty-reducing adversarial training method.

### 5.1 ENTROPY MINIMIZATION FOR CP-EFFICIENCY

The prediction set size is closely related to the entropy of prediction distribution, as both quantities reflect the prediction uncertainty of a model. A more uniform categorical distribution has higher uncertainty, which is reflected in its higher entropy. Fig. 3 visualizes the kernel density estimation (KDE) (Rosenblatt, 1956; Parzen, 1962) of entropy values calculated with adversarial test samples

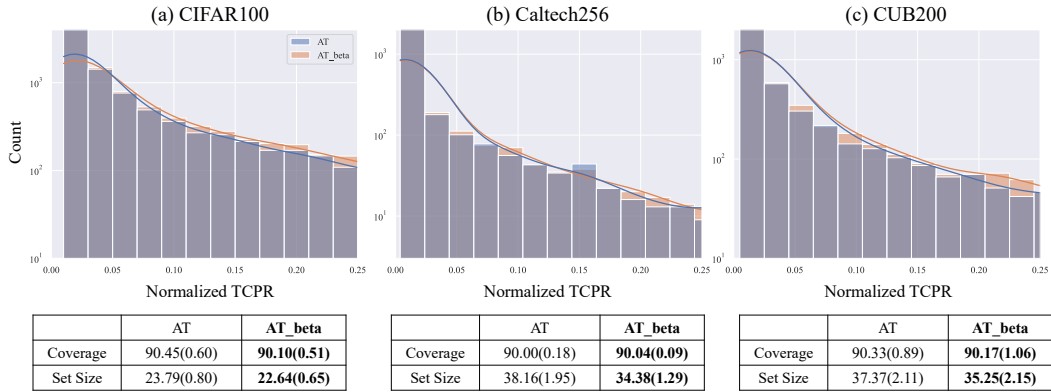

| | AT | AT_beta |
|---|---|---|
| Coverage | 90.45(0.60) | **90.10(0.51)** |
| Set Size | 23.79(0.80) | **22.64(0.65)** |

| | AT | AT_beta |
|---|---|---|
| Coverage | 90.00(0.18) | **90.04(0.09)** |
| Set Size | 38.16(1.95) | **34.38(1.29)** |

| | AT | AT_beta |
|---|---|---|
| Coverage | 90.33(0.89) | **90.17(1.06)** |
| Set Size | 37.37(2.11) | **35.25(2.15)** |

Figure 5: **Top**: The histogram and kernel density estimation of normalized TCPR on adversarial test sets. **Bottom**: The coverage and prediction set size of AT and AT-Beta. AT-Beta pushes the TCPR distribution towards the promising region and improves CP-efficiency.

on three datasets. It is evident that TRADES and MART learn models with predictive distributions that have higher entropy values than AT, thus increasing the prediction set size comparatively.

To decrease the prediction set size of AT, we add an entropy minimization term to the loss function,

$$\ell_{\text{EM}}(f_\theta(x_i), y_i) = -\sum_j^K y_{ij} \log(f_\theta(x_i)_j) + H(f_\theta(x_i)), \tag{4}$$

where the regularization is the entropy function $H(f_\theta(x_i)) = -\sum_j^K f_\theta(x_i)_j \log(f_\theta(x_i)_j)$. The AT scheme with entropy minimization (EM) is denoted as AT-EM. This entropy term is the same as the entropy minimization in semi-supervised learning (Grandvalet & Bengio, 2004). However, note that our work is the first to use entropy minimization in adversarial training for improving CP-efficiency. Fig. 3 also shows the KDE of entropy values on adversarial test sets using AT-EM. The reduction in predictive entropy effectively leads to a substantial decrease in the prediction set size of AT-EM.

## 5.2 BETA WEIGHTING FOR CP-EFFICIENCY

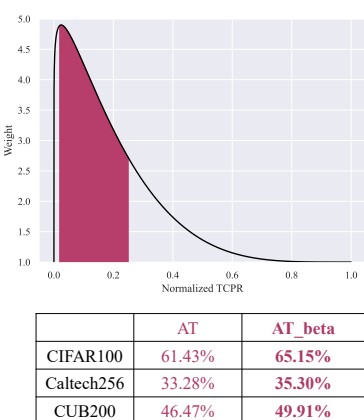

| | AT | AT_beta |
|---|---|---|
| CIFAR100 | 61.43% | **65.15%** |
| Caltech256 | 33.28% | **35.30%** |
| CUB200 | 46.47% | **49.91%** |

Figure 4: Top: The Beta distribution density function used in our experiment. Bottom: The ratio of $p_{\text{Beta}}(\hat{r}_i)$ within the promising TCPR region (red) to the summation of all $p_{\text{Beta}}(\hat{r}_i)$'s.

The second factor that controls prediction set sizes is the distribution of True Class Probability Ranking (TCPR) on the test dataset. The TCPR is defined as the ranking of a sample $x$'s ground-truth class probability among the whole predictive probability. In equation, we sort $\pi$ with the descending order into $\hat{\pi}$,

$$\hat{\pi} = \{\pi_{(1)}, \cdots, \pi_{(K)}\}, \tag{5}$$

where $\pi_{(j)} \geq \pi_{(j+1)}, \forall j = 1, \cdots, K-1$, and $(j)$ is the sorted index. TCPR is the index $j$ in $\hat{\pi}$ corresponding to the ground-truth label $y$, i.e., $Sort(y) = j$.

The TCPR matters to the prediction set size as we observe that a model with higher robust accuracy does not necessarily have a smaller prediction set size as shown in Tab. 1. This discovery indicates that improving Top-1 accuracy, i.e., the percentage of samples with TCPR=1, is not enough to learn a CP-efficient model. In addition, the model capacity might be not strong enough to fit all the adversarial training data or achieve 100% adversarial training accuracy as a result of strong adversary and high task complexity, such as a large number of classes. For instance, on CIFAR100, the robust accuracy on training data using a pre-trained ResNet50 is only 46.1%.

Motivated by this observation, we propose to use a Beta distribution density function (Fig. 4) to weight the loss samples so that the TCPR distribution shifts towards the lower TCPR region. This design embodies our insight that the training should focus on samples with *promising* TCPR's, whose TCPR's are not 1 and also not too large, because TCPR of 1 means the sample is correctly classified and a large TCPR means the sample is too difficult. Those samples with promising TCPR's are important to control prediction set sizes as they are the *majority* of the dataset and thus largely affect the averaged prediction set sizes. Note that the Beta weighting is fundamentally different

from Focal Loss (Lin et al., 2017), which focuses on hard examples, i.e., samples with higher TCPR are weighted higher. Our experiment demonstrates that focusing hard samples leads to inferior CP-effiency.

Thus, we propose an importance weighting scheme based on Beta distribution density function of TCPR to learn a CP-efficient model. Let the TCPR of sample $\tilde{x}_i$ be $r_i \in [K]$ and the normalized TCPR be $\hat{r}_i \in [0, 1]$. We use the Beta distribution density function, e.g., Fig. 4, to give an importance weight to sample $\tilde{x}_i$. The Beta distribution density is

$$p_{\text{Beta}}(z; a, b) = \frac{\Gamma(a + b)}{\Gamma(a)\Gamma(b)} \cdot (z)^{a-1} \cdot (1 - z)^{b-1}, \tag{6}$$

where $\Gamma(a)$ is the Gamma function. To enforce the model to focus on samples with promising TCPR's, we use the Beta distribution with $a = 1.1$ and $b \in \{3.0, 4.0, 5.0\}$. When $a = 1.1$ and $b = 5.0$, we have the Beta weighting function shown in Fig. 4. The objective function of Beta-weighting AT is

$$\ell_{\text{Beta}}(f_\theta(x_i), y_i) = -p_{\text{Beta}}(\hat{r}_i; a, b) \cdot \sum_{j}^{K} y_{ij} \log(f_\theta(x_i)_j) \tag{7}$$

We name this Beta distribution based importance weighting scheme in AT as AT-Beta. The TCPR distributions of adversarial test samples when training with AT and AT-Beta are shown in Fig. 5, where AT-Beta has more density around the promising region. The CP performance is also shown below each plot, which validates the effectiveness of AT-Beta in improving CP-efficiency. In the table of Fig. 4, we show the ratio of $p_{\text{Beta}}(\hat{r}_i)$ within the promising TCPR region (red) to the summation of all $p_{\text{Beta}}(\hat{r}_i)$'s, where AT-Beta has substantially more promising weight values than AT.

In summary, the proposed AT-UR consists of two methods, AT-Beta and AT-EM. It also contains the combination of the two methods, i.e.,

$$\ell_{\text{Beta-EM}}(f_\theta(x_i), y_i) = -p_{\text{Beta}}(\hat{r}_i; a, b) \cdot \sum_{j}^{K} y_{ij} \log(f_\theta(x_i)_j) + H(f_\theta(x_i)), \tag{8}$$

denoted as AT-Beta-EM. We test the three variants of AT-UR in our experiment and observe that different image classification tasks need different versions of AT-UR.

### 5.3 THEORETICAL ANALYSIS ON BETA WEIGHTING

We next give our theoretical analysis on the Beta weighting, which shows that the generalization error of Beta weighting is potentially improved compared with uniform weighting. We drop the subscript $\theta$ for $f_\theta$ to lighten the notation. Note that we leave the full proof in Appendix D.

**Importance Weighting (IW) Algorithm.** IW assigns importance weight $\omega(x, y)$ to each sample $(x, y) \in \mathcal{D}_{\text{tr}}$ such that $\omega(x, y)$ is directly determined by TCPR $\hat{r}$. Analogous to the empirical risk $\widehat{R}(f)$, we define the *IW empirical risk* with weights $\omega(x, y)$ for $f$ as follows

$$\widehat{R}_\omega(f) = \frac{1}{m} \sum_{i=1}^{m} \omega(x_i, y_i) \cdot \ell(f(x_i), y_i). \tag{9}$$

It is worth noting that restricting $\omega(x_i, y_i) = 1$ as a special case for all data samples reduces $\widehat{R}_\omega(f)$ to $\widehat{R}(f)$. With some configurations of $\omega$ under the general IW framework, the generalization error bound can be tightened compared to the standard case.

We strategically design a group-wise IW approach that groups data into $K$ disjoint subsets according to the rank of their true labels, and assign the same weight to a group of data. For a sample $(x, y)$, the importance weight is $\omega(x, y) = p_{\text{Beta}}(\hat{r}(x, y); a, b)$.

**Lemma 1.** *(Generalization error bound of IW empirical risk, Theorem 1 in Cortes et al. (2010)) Let $M = \sup_{(x,y) \in \mathcal{X} \times \mathcal{Y}} \omega(x, y)$ denote the infinity norm of $\omega$ on the domain. For given $f \in \mathcal{F}$ and $\delta > 0$, with probability at least $1 - \delta$, the following bound holds:*

$$R(f) - \widehat{R}_\omega(f) \leq \frac{2M \log(1/\delta)}{3m} + \sqrt{\frac{2d_2(\mathcal{P} \| \frac{\mathcal{P}}{\omega}) \log(1/\delta)}{m}}, \tag{10}$$

*where $d_2(\mathcal{P} \| \mathcal{Q}) = \int_x \mathcal{P}(x) \cdot \frac{\mathcal{P}(x)}{\mathcal{Q}(x)} dx$ is the base-2 exponential for Rényi divergence of order 2 between two distributions $\mathcal{P}$ and $\mathcal{Q}$ and $m$ is the number of training samples.*

| Dataset | CIFAR10 | | CIFAR100 | | Caltech256 | | CUB200 | |
|---|---|---|---|---|---|---|---|---|
| Metric | Cvg | PSS | Cvg | PSS | Cvg | PSS | Cvg | PSS |
| AT | 90.55(0.51) | 3.10(0.07) | 90.45(0.59) | 23.79(0.80) | 91.35(0.85) | 43.20(2.11) | 90.33(0.89) | 37.37(2.11) |
| AT-EM* | 90.39(0.48) | **3.05(0.05)** | 90.35(0.82) | **22.05(1.02)** | 91.09(0.79) | 41.42(2.52) | 90.08(1.10) | 34.77(2.60) |
| AT-Beta* | 90.46(0.51) | 3.11(0.07) | 90.10(0.51) | 22.64(0.65) | 90.20(0.84) | **35.39(2.66)** | 90.17 (1.06) | 35.25(2.15) |
| AT-Beta-EM* | 90.65(0.62) | 3.10(0.08) | 90.40(0.60) | 22.53(0.91) | 90.81(1.00) | 36.17(3.73) | 90.31(0.84) | **33.10(1.74)** |
| FAT | 90.69(0.61) | 3.16(0.07) | 90.41(0.67) | 23.54(0.81) | 90.70(0.77) | 41.52(2.43) | 90.50(1.17) | 39.43(2.88) |
| FAT-EM* | 90.54(0.68) | 3.06(0.06) | 90.00(0.82) | 23.47(2.71) | 90.55(0.79) | 39.72(2.49) | 89.89(0.919) | 35.51(2.06) |
| FAT-Beta* | 90.47(0.51) | 3.16(0.08) | 90.22(0.47) | 23.15(0.71) | 89.90(0.70) | 34.72(2.25) | 89.92(0.84) | 35.46(1.71) |
| FAT-Beta-EM* | 90.71(0.61) | **3.04(0.07)** | 90.36(0.50) | **22.28(0.63)** | 90.41(0.61) | **33.59(2.75)** | 89.88(0.91) | **34.35(1.68)** |
| TRADES | 90.72(0.62) | 3.31(0.09) | 90.35(0.57) | 27.60(0.97) | 90.82(0.81) | 44.80(3.42) | 90.38(0.76) | 52.18(2.60) |
| TRADES-EM* | 90.54(0.40) | **3.16(0.05)** | 90.36(0.71) | **26.76(1.00)** | 90.68(0.87) | **38.83(3.78)** | 90.05(0.76) | **44.96(2.75)** |
| TRADES-Beta* | 90.41(0.56) | 3.30(0.09) | 90.14(0.85) | 27.22(1.42) | 90.48(0.70) | 38.94(2.74) | 89.83(0.84) | 49.63(2.59) |
| TRADES-Beta-EM* | 90.01(0.40) | 3.21(0.06) | 90.54(0.24) | 26.55(0.35) | 90.52(0.74) | 39.83(3.02) | 90.16(1.12) | 48.54(2.58) |

Table 2: Comparison of AT baselines and the proposed AT-UR variants denoted with *. The average coverage (Cvg) and prediction set size (PSS) are presented, along with the standard deviation in parentheses. The most CP-efficient method is highlighted in bold.

**Theorem 1.** *(Beta weighting preserves generalization error bound.) Suppose* $\mathbb{P}_{(x,y)\sim\mathcal{P}}\{\hat{r}(x,y) = k\} = \frac{k^{-c}}{\sum_{k'=1}^{K}(k')^{-c}}$ *is a polynomially decaying function with* $c = \max\{K^{-\alpha}, \frac{b\ln(a)+1}{\ln(K)} + 2 - \alpha\}$ *for* $\alpha \geq 0$. *Beta weighting improves generalization error bound compared with ERM.*

**Remark.** Theorem 1 shows that the Beta weighting approach guarantees improved generalization error bound, which is beneficial to ensure the desirable accuracy for prediction. Meanwhile, the Beta-based IW strategy focuses on penalizing the data samples whose prediction set size is moderately large (e.g., 10-20 labels included out of 100+ class labels, see experiments).

## 6 EXPERIMENT

We first give the details of our experimental setting and then present the main empirical result.

### 6.1 EXPERIMENTAL SETTING

**Model.** We use the adversarially pre-trained ResNet50 (He et al., 2016; Salman et al., 2020) with $l_\infty$ norm and an attack budget $\epsilon_{pt} = 4/255$ in all experiment of our paper. The reason is that, besides testing on CIFAR10/100, we also test on more challenging datasets such as Caltech256 and CUB200, on which an adversarially pre-trained model is shown to be much more robust than random initialized weights (Liu et al., 2023).

**Dataset.** Four datasets are used to evaluate our method, i.e., CIFAR10, CIFAR100 (Krizhevsky et al., 2009), Caltech-256 (Griffin et al., 2007) and Caltech-UCSD Birds-200-2011 (CUB200) (Wah et al., 2011). CIFAR10 and CIFAR100 contain low-resolution images of 10 and 100 classes, where the training and validation sets have 50,000 and 10,000 images respectively. Caltech-256 has 30,607 high-resolution images and 257 classes, which is split into training and validation set using a 9:1 ratio. CUB200 also contains high-resolution bird images for fine-grained image classification, with 200 classes, 5,994 training images and 5,794 validation images.

**Training and Adversarial Attack.** In all adversarial training of this paper, we generate adversarial perturbations using PGD attack. The PGD attack has 10 steps, with stepsize $\lambda = 2/255$ and attack budget $\epsilon = 8/255$. The batch size is set as 128 and the training epoch is 60. We divide the learning rate by 0.1 at the 30th and 50th epoch. We use PGD attack with 100 steps for evaluation in this paper. The stepsize and attack budget is the same as in adversarial training, i.e., $\lambda = 2/255$ and $\epsilon = 8/255$. See more training details in Appendix B.

**Conformal Prediction Setting.** We fix the training set in our experiment and randomly split the original test set into calibration and test set with a ratio of 1:4 for conformal prediction. For each AT method, we repeat the training for three trials with three different seeds and repeat the calibration-test splits five times, which produces 15 trials for our evaluation. The mean and standard deviation of coverage and prediction set size of 15 trials are reported. If not specified, we use APS (Romano et al., 2020) as the CP method in our paper as the performance of APS is more stable than RAPS, as shown in Fig. 2. The target coverage is set as 90% following existing literature in CP (Romano et al., 2020; Angelopoulos et al., 2020; Ghosh et al., 2023). We use the same adversarial attack setting as in Gendler et al. (2021), i.e., both calibration and test samples are attacked with the same adversary. We discuss the limitation of this setting in the conclusion section.

**Baselines.** We use AT (Madry et al., 2018), Fair-AT (FAT) (Xu et al., 2021) and TRADES (Zhang et al., 2019) as the baseline and test the performance of the two proposed uncertainty-reducing methods with the three baselines. AT and TRADES are the most popular adversarial training methods and FAT reduces the robustness variance among classes, which could reduce the prediction set size, which is validated by our experiment. Note that we only reports the performance of CP, i.e., cov-

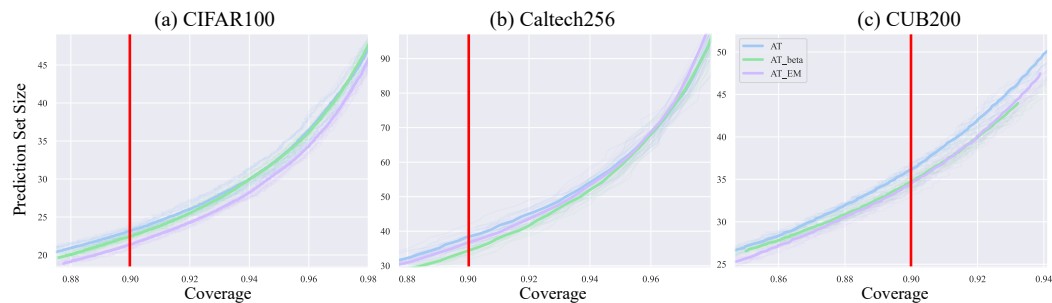

Figure 6: The CP curve of coverage versus prediction set size. Each point on the curve is obtained by adjusting the threshold $\hat{\tau}_{\text{cal}}$. We plot 15 CP curves (opaque line) and their average (solid line) for each method. The red vertical line indicates the operating point for 90% coverage.

erage and prediction set size, in the main paper as the main target of our paper is to improve CP efficiency.

## 6.2 EXPERIMENTAL RESULTS

**Efficacy of AT-UR in reducing prediction set sizes.** The coverage and prediction set size of all tested methods under the PGD100 attack are shown in Tab. 2. The proposed AT-UR methods effectively reduce the prediction set size when combined with the three AT baselines on four datasets, validating our intuition on the connection between the two factors, i.e., predictive entropy and the distribution of TCPR, and PSS's. There are two phenomena worth noting. First, the EM version of AT-UR generally works better than Beta and Beta-EM on CIFAR10 and CIFAR100, where EM achieves the lowest prediction set size in AT and TRADES. Second, on the two challenging datasets, Caltech256 and CUB200, the Beta-EM variant of AT-UR can improve either EM or Beta, e.g., AT on CUB and FAT on all four datasets. Based on the two observations, we recommend that for high-complexity classification tasks, the Beta-EM is the first choice if one needs to train an adversarially robust and also CP-efficient model. Note that although the Top-1 accuracy of our method (Appendix C) is decreased compared to baselines, the main target of our method is to improve CP efficiency.

**Coverage-PSS curve visualization.** To visualize the effect of AT-UR more comprehensively, we plot the CP curve by adjusting the threshold $\hat{\tau}_{\text{cal}}$ to get different points on the curve of coverage versus prediction set size. Fig. 6 shows the CP curve of AT, AT-Beta and AT-EM on three datasets. It demonstrates that AT-UR achieves a reduced prediction set size compared to the AT baseline, not only at 90% coverage , but also over a wide range of coverage values.

## 6.3 ABLATION STUDY

**Does Focal loss improve CP-efficiency?** We consider using a power function $\hat{r}_i^{\eta}$ as in focal loss (Lin et al., 2017) to generate loss weights and test the CP performance of AT-Focal. We set $\eta = 0.5$ based on a hyperparameter search from $\{0.1, 0.5, 1.0, 2.0\}$. AT-Focal forces the model to focus on hard samples, contrary to our AT-Beta which focuses on promising samples. The averaged coverage and PSS of AT-Focal on CIFAR100 and Caltech256 are (90.50, 27.24) and (91.38, 48.35) respectively, which is far worse than the AT baseline of (90.45, 23.79) and (91.35, 43.20). This result corroborates that promising samples are crucial for improving CP-efficiency instead of hard samples.

**What is the difference between label smoothing and AT-EM?** The formulation of AT-EM is similar to the formulation of label smoothing (Müller et al., 2019), if we combine the log term in (4). However, label smoothing and AT-EM train the model into two different directions: the former increases the prediction entropy (by smoothing the label probabilities to be more uniform), while the latter decreases the prediction entropy. We validate this argument on Caltech256 and find that label smoothing makes the CP-efficiency much worse than the AT baseline, with an averaged coverage and prediction set size of (90.22, 46.39), compared to (91.35, 43.20) of AT.

## 7 CONCLUSION

This paper first studies the pitfalls of CP under adversarial attacks and thus underscores the importance of AT when using CP in an adversarial environment. Then we unveil the compromised CP-efficiency of popular AT methods and propose to design uncertainty-reducing AT for CP-efficiency based on our thorough empirical study on two factors affecting the prediction set size. Our experiment validates the effectiveness of the proposed AT-UR on four datasets when combined with three AT baselines. A common limitation shared by this study and Gendler et al. (2021) is the assumption that the adversarial attack is known, enabling the calibration set to be targeted by the same adversary as the test set. In future research, we will alleviate this constraint by exploring CP within an adversary-agnostic context.

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

# A ADAPTIVE PREDICTION SETS (ROMANO ET AL., 2020)

We introduce one example of prediction set function, i.e., APS conformal prediction used in our experiment. Assume we have the prediction distribution $\pi(x) = f_\theta(x)$ and order this probability vector with the descending order $\pi_{(1)}(x) \geq \pi_{(2)}(x) \geq \ldots \geq \pi_{(K)}(x)$. We first define the following generalized conditional quantile function,

$$Q(x; \pi, \tau) = \min\{k \in \{1, \ldots, K\} : \pi_{(1)}(x) + \pi_{(2)}(x) + \ldots + \pi_{(k)}(x) \geq \tau\}, \qquad (11)$$

which returns the class index with the generalized quantile $\tau \in [0, 1]$. The function $\mathcal{S}$ can be defined as

$$\mathcal{S}(x, u; \pi, \tau) = \begin{cases} \text{`}y\text{' indices of the } Q(x; \pi, \tau) - 1 \text{ largest } \pi_y(x), & \text{if } u \leq V(x; \pi, \tau), \\ \text{`}y\text{' indices of the } Q(x; \pi, \tau) \text{ largest } \pi_y(x), & \text{otherwise,} \end{cases} \qquad (12)$$

where

$$V(x; \pi, \tau) = \frac{1}{\pi_{(Q(x;\pi,\tau))}(x)} \left\lceil \sum_{k=1}^{Q(x;\pi,\tau)} \pi_{(k)}(x) - \tau \right\rceil.$$

It has input $x$, $u \in [0, 1]$, $\pi$, and $\tau$ and can be seen as a generalized inverse of Equation 11.

On the calibration set, we compute a generalized inverse quantile conformity score with the following function,

$$E(x, y, u; \pi) = \min \{\tau \in [0, 1] : y \in \mathcal{S}(x, u; \pi, \tau)\}, \qquad (13)$$

which is the smallest quantile to ensure that the ground-truth class is contained in the prediction set $\mathcal{S}(x, u; \pi, \tau)$. With the conformity scores on calibration set $\{E_i\}_{i=1}^{n_c}$, we compute the $\lceil (1 - \alpha)(1 + n_c)\rceil$th largest value in the score set as $\hat{\tau}_{\text{cal}}$. During inference, the prediction set is generated with $\mathcal{S}(x^*, u; \pi^*, \hat{\tau}_{\text{cal}})$ for a novel test sample $x^*$.

# B MORE EXPERIMENTAL DETAILS

**APS Setting.** We use the default setting of APS specified in the official code of Angelopoulos et al. (2020), i.e., first use temperature scaling (Platt et al., 1999; Guo et al., 2017) to calibrate the prediction distribution then compute the generalized inverse quantile conformity score to perform the calibration and conformal prediction.

**Hyperparameter and Baseline Setting.** As mentioned in the main paper, we use $a = 1.1$ and search $b$ from the discrete set $\{2.0, 3.0, 4.0, 5.0\}$ in Beta distribution since the parameter combinations perform well in our pilot study and satisfy the goal of focusing on promising samples. The learning rate and weight decay of AT, FAT and TRADES are determined by grid search from {1e-4,3e-4,1e-3,3e-3,1e-2} and {1e-3,1e-4,1e-5} respectively. We compute the class weight for FAT using the output of a softmax function with error rate of each class as input. The temperature in the softmax function is set as 1.0. For TRADES, we follow the default setting $\beta = 6.0$ for the KL divergence term (Zhang et al., 2019). Our AT-UR method also determines the learning rate and weight decay using the grid search with the same mentioned grid. For TRADES, we weight both the cross-entropy loss and KL divergence loss with the Beta density function based on TCPR.

**CP Curve.** The CP curve in Fig. 6 is obtained by using different threshold values, for instance, using the linspace function in numpy (Harris et al., 2020) with `np.linspace(0.9,1.1,200)` $\times \hat{\tau}_{\text{cal}}$ generates 200 different (coverage, PSS) points.

# C MORE EXPERIMENTAL RESULTS

Note that this paper uses CP as the inference method to achieve a coverage guarantee, which is orthogonal to the Top-1 inference method. Thus, Top-1 accuracy is not a relevant metric in the context of CP inference. Nevertheless, we show the Top-1 accuracy of tested methods in Tab. 4. Using AT-UR generally worsens the Top-1 accuracy, especially for TRADES. However, note that using TRADES-Beta-EM can improve the Top-1 robust accuracy of TRADES-Beta on CIFAR10 and TRADES-EM on Caltech256. This result again confirms the observation that Top-1 accuracy is not necessarily correlated with CP-efficiency. We use the AutoAttack (AA) Croce & Hein (2020)

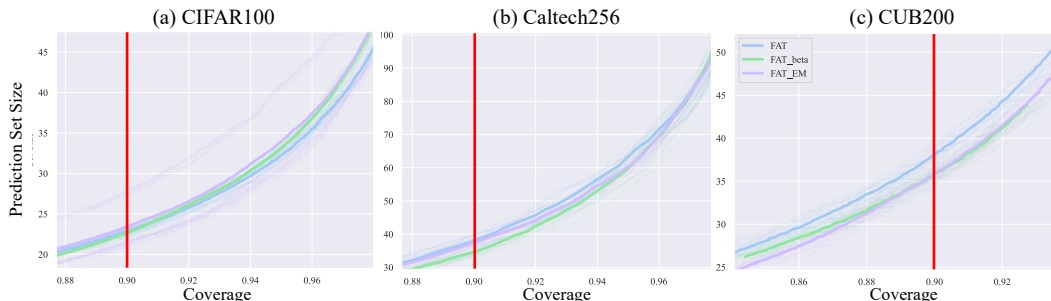

Figure 7: The CP curve of coverage versus prediction set size using FAT.

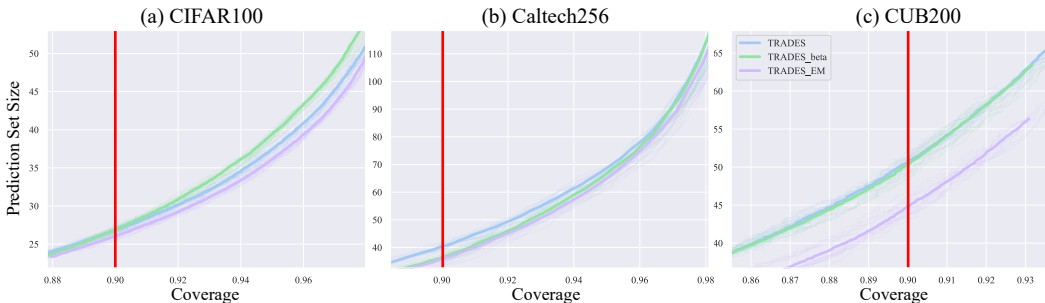

Figure 8: The CP curve of coverage versus prediction set size using TRADES.

to evaluate our method with AT, as AA is the most popular adversarial attack in literature due to its effectiveness. See Tab. 5 for the result. Under AA, the performance of our approach is also substantially better than the AT baseline. Interestingly, the robust accuracy under AA drops while the prediction set size reduced (CP efficiency is improved).

Fig. 7 and Fig. 8 shows the CP curve of FAT and TRADES when they are combined with EM and Beta on three datasets. It demonstrates that the CP-efficiency is also improved when using FAT and TRADES as in the experiment using AT. In most cases (5 out of 6), AT-UR (either EM or Beta) has a lower PSS than the corresponding baseline within a large range of coverage.

## D    PROOF OF THEOREM 1

**Theorem 2.** *(Theorem 1 restated, Beta weighting preserves generalization error bound.) Suppose* $\mathbb{P}_{(x,y)\sim\mathcal{P}}\{\hat{r}(x,y) = k\} = \frac{k^{-c}}{\sum_{k'=1}^{K}(k')^{-c}}$ *is a polynomially decaying function with* $c = \max\{K^{-\alpha}, \frac{b\ln(a)+1}{\ln(K)} + 2 - \alpha\}$ *for* $\alpha \geq 0$. *Beta weighting improves generalization error bound compared with ERM.*

*Proof.* (of Theorem 1)

The key idea to prove Theorem 1 is to show $d_2(\mathcal{P}||\mathcal{P}/\omega) \leq d_2(\mathcal{P}||\mathcal{P}) = 1$ (recall $d_2$ is the base-2 exponential for Rényi divergence of order 2, as in Lemma 1), which implies that Beta weighting gives tighter generalization error bound than ERM.

| Dataset | CIFAR10 | | CIFAR100 | | Caltech256 | | CUB200 | |
|---|---|---|---|---|---|---|---|---|
| Metric | Cvg | NPSS | Cvg | NPSS | Cvg | NPSS | Cvg | NPSS |
| AT | 90.55(0.51) | 31.0(0.7) | 90.45(0.59) | 23.79(0.80) | 91.35(0.85) | 16.8(0.8) | 90.33(0.89) | 18.7(1.1) |
| AT-EM* | 90.39(0.48) | **30.5(0.5)** | 90.35(0.82) | **22.05(1.02)** | 91.09(0.79) | 16.1(1.0) | 90.08(1.10) | 17.4(1.3) |
| AT-Beta* | 90.46(0.51) | 31.1(0.7) | 90.10(0.51) | 22.64(0.65) | 90.20(0.84) | **13.8(1.0)** | 90.17(1.06) | 17.6(1.1) |
| AT-Beta-EM* | 90.65(0.62) | 31.0(0.8) | 90.40(0.60) | 22.53(0.91) | 90.81(1.00) | 14.1(1.4) | 90.31(0.84) | **16.5(0.9)** |
| FAT | 90.69(0.61) | 31.6(0.7) | 90.41(0.67) | 23.54(0.81) | 90.70(0.77) | 16.2(0.9) | 90.50(1.17) | 19.7(1.4) |
| FAT-EM* | 90.54(0.68) | 30.6(0.6) | 90.00(0.82) | 23.47(2.71) | 90.55(0.79) | 15.5(1.0) | 89.89(0.919) | 17.8(1.0) |
| FAT-Beta* | 90.47(0.51) | 31.6(0.8) | 90.22(0.47) | 23.15(0.71) | 89.90(0.70) | 13.5(0.9) | 89.92(0.84) | 17.7(0.9) |
| FAT-Beta-EM* | 90.71(0.61) | **30.4(0.7)** | 90.36(0.50) | **22.28(0.63)** | 90.41(0.61) | **13.1(1.1)** | 89.88(0.91) | **17.2(0.8)** |
| TRADES | 90.72(0.62) | 33.1(0.9) | 90.35(0.57) | 27.60(0.97) | 90.82(0.81) | 17.4(1.3) | 90.38(0.76) | 26.1(1.3) |
| TRADES-EM* | 90.54(0.40) | **31.6(0.5)** | 90.36(0.71) | **26.76(1.00)** | 90.68(0.87) | **15.1(1.5)** | 90.05(0.76) | **22.5(1.4)** |
| TRADES-Beta* | 90.41(0.56) | 33.0(0.9) | 90.14(0.85) | 27.22(1.42) | 90.48(0.70) | 15.1(1.1) | 89.83(0.84) | 24.8(1.3) |
| TRADES-Beta-EM* | 90.01(0.40) | 32.1(0.6) | 90.54(0.24) | 26.55(0.35) | 90.52(0.74) | 15.5(1.2) | 90.16(1.12) | 24.3(1.3) |

Table 3: Comparison of AT baselines and the proposed AT-UR variants denoted with *. The average coverage (Cvg) and prediction set size normalized by the class number (NPSS, %) are presented.

| Dataset | CIFAR10 | | CIFAR100 | | Caltech256 | | CUB200 | |
|---|---|---|---|---|---|---|---|---|
| Metric | Std. Acc. | Rob. Acc. | Std. Acc. | Rob. Acc. | Std. Acc. | Rob. Acc. | Std. Acc. | Rob. Acc. |
| AT | 89.76(0.15) | 50.17(0.91) | 68.92(0.38) | 28.49(1.14) | 75.28(0.51) | 47.53(0.67) | 65.36(0.27) | 26.29(0.45) |
| AT-EM* | 90.02(0.10) | 48.92(0.39) | 68.39(0.51) | 28.33(0.73) | 74.62(0.22) | 46.23(0.44) | 64.75(0.33) | 25.60(0.25) |
| AT-Beta* | 89.81(0.22) | 47.50(0.78) | 68.50(0.28) | 28.04(0.57) | 74.66(0.54) | 45.40(0.59) | 64.62(0.17) | 25.57(0.41) |
| AT-Beta-EM* | 90.00(0.06) | 46.69(0.71) | 68.45(0.35) | 27.20(1.08) | 74.71(0.36) | 44.88(0.60) | 64.44(0.22) | 25.32(0.38) |
| FAT | 89.96(0.25) | 49.12(0.70) | 68.80(0.38) | 28.97(0.53) | 75.20(0.33) | 47.09(0.70) | 65.01(0.19) | 25.21(0.57) |
| FAT-EM* | 90.19(0.07) | 48.31(0.80) | 68.76(0.39) | 25.84(3.54) | 74.59(0.20) | 45.53(0.70) | 65.13(0.25) | 24.92(0.27) |
| FAT-Beta* | 90.07(0.16) | 47.09(0.50) | 68.58(0.33) | 27.90(0.56) | 74.00(0.39) | 45.12(0.90) | 64.38(0.25) | 24.76(0.26) |
| FAT-Beta-EM* | 89.86(0.06) | 48.61(0.13) | 67.95(0.24) | 28.06(0.24) | 74.78(0.10) | 45.75(0.48) | 64.32(0.21) | 23.57(0.14) |
| TRADES | 87.31(0.27) | 53.07(0.23) | 62.83(0.33) | 32.07(0.20) | 69.57(0.25) | 47.07(0.37) | 58.16(0.38) | 27.82(0.23) |
| TRADES-EM* | 86.68(0.06) | 52.71(0.26) | 57.03(0.31) | 30.29(0.25) | 57.17(0.39) | 39.56(0.59) | 45.50(5.81) | 22.26(2.26) |
| TRADES-Beta* | 89.81(0.22) | 47.50(0.78) | 62.61(0.36) | 30.20(0.31) | 70.96(0.25) | 46.74(0.23) | 57.72(0.24) | 23.49(0.21) |
| TRADES-Beta-EM* | 86.99(0.10) | 51.85(0.23) | 62.13(0.34) | 30.52(0.20) | 69.44(0.25) | 46.24(0.39) | 56.03(0.15) | 22.90(0.32) |

Table 4: Top-1 clean and robust accuracy comparison of AT baselines and the proposed AT-UR variants.

First, we derive the following equivalent formulations.

$$d_2(\mathcal{P}||\mathcal{P}/\omega) = \int_{(x,y)} \mathcal{P}(x,y) \cdot \omega(x,y) d(x,y) = \int_{(x,y)} \mathcal{P}(x,y) \cdot p_{\text{Beta}}(\hat{r}(x,y)/K; a, b) d(x,y)$$

$$= \int_{(x,y)} \mathcal{P}(x,y) \left( \sum_{k=1}^{K} \mathbb{I}[\hat{r}(x,y) = 1] \right) \cdot p_{\text{Beta}}(\hat{r}(x,y)/K; a, b) d(x,y)$$

$$= \sum_{k=1}^{K} \int_{(x,y)} \mathcal{P}(x,y) \cdot \mathbb{I}[\hat{r}(x,y) = 1] \cdot p_{\text{Beta}}(\hat{r}(x,y)/K; a, b) d(x,y)$$

$$= \sum_{k=1}^{K} \int_{(x,y)} \mathcal{P}(x,y) \cdot \mathbb{I}[\hat{r}(x,y) = 1] \cdot p_{\text{Beta}}(k/K; a, b) d(x,y)$$

$$= \sum_{k=1}^{K} \underbrace{\mathbb{P}_{(x,y)\sim\mathcal{P}}\{\hat{r}(x,y) = k\}}_{=p_k} \cdot p_{\text{Beta}}(k/K; a, b).$$

| Dataset | CIFAR10 | | CIFAR100 | | Caltech256 | | CUB200 | |
|---|---|---|---|---|---|---|---|---|
| Metric | Cvg | PSS | Cvg | PSS | Cvg | PSS | Cvg | PSS |
| AT | 93.25(0.45) | 2.54(0.04) | 91.99 (0.61) | 14.29(0.59) | 94.35(0.81) | 23.73(1.68) | 91.87(0.90) | 17.75(0.71) |
| AT-EM* | 92.36(0.53) | **2.45(0.04)** | 91.87(0.61) | 13.29(0.49) | 93.41(0.58) | 21.19(1.46) | 91.26(0.57) | **16.47(0.61)** |
| AT-Beta* | 91.96(0.39) | 2.50(0.04) | 91.24(0.69) | **11.61(0.40)** | 93.52(0.73) | **18.54(1.32)** | 91.37(0.75) | 16.56(0.76) |
| AT-Beta-EM* | 92.06(0.44) | 2.50(0.04) | 91.13(0.63) | 11.78(0.47) | 93.50(0.69) | 18.56(1.32) | 91.93(0.68) | 16.67(0.58) |

Table 5: Comparison of AT and the proposed AT-UR variants denoted with * under AutoAttack Croce & Hein (2020).

Suppose $p_k = \frac{k^{-c}}{\sum_{k'=1}^{K}(k')^{-c}}$ a polynomially decaying function of $k$ for $c \geq 0$.

$$p_k \cdot p_{\text{Beta}}(k/K)$$

$$= \frac{k^{-c}}{\sum_{k'=1}^{K}(k')^{-c}} \cdot \frac{\Gamma(a+b)}{\Gamma(a)\Gamma(b)} \cdot (\frac{k}{K})^{a-1} \cdot (1 - \frac{k}{K})^{b-1}$$

$$= \frac{K^{-c}}{K^{-c}} \cdot \frac{k^{-c}}{\sum_{k'=1}^{K}(k')^{-c}} \cdot \frac{(a+b-1)!}{(a-1)!(b-1)!} \cdot (\frac{k}{K})^{a-1} \cdot (1 - \frac{k}{K})^{b-1}$$

$$= \frac{K^{-c}}{\sum_{k'=1}^{K}(k')^{-c}} \cdot \frac{k^{-c}}{K^{-c}} \cdot \frac{(a-c+b-1)! \cdot \prod_{i=a-c+b}^{a+b-1} i}{(a-c-1)!(b-1)! \cdot \prod_{i=a-c}^{a-1} i} \cdot (\frac{k}{K})^{a-1} \cdot (1 - \frac{k}{K})^{b-1}$$

$$= \frac{K^{-c}}{\sum_{k'=1}^{K}(k')^{-c}} \cdot \frac{\Gamma(a-c+b)}{\Gamma(a-c)\Gamma(b)} \cdot \prod_{i=1}^{c} \frac{a+b-c-1+i}{a-c-1+i} \cdot (\frac{k}{K})^{a-c-1} \cdot (1 - \frac{k}{K})^{b-1}$$

$$= \underbrace{\frac{K^{-c}}{\sum_{k'=1}^{K}(k')^{-c}}}_{=A} \cdot \underbrace{\prod_{i=1}^{c} (1 + \frac{b}{a-c-1+i})}_{=B} \cdot \underbrace{\frac{\Gamma(a-c+b)}{\Gamma(a-c)\Gamma(b)} \cdot (\frac{k}{K})^{a-c-1} \cdot (1 - \frac{k}{K})^{b-1}}_{=p_{\text{Beta}}(k/K;a-c,b)}$$

where term $A$ can be bounded as follows

$$\frac{K^{-c}}{\sum_{k=1}^{K} k^{-c}} \leq \frac{K^{-c}(c-1)}{1 - (K+1)^{-(c-1)}} \leq cK^{-c}, \quad c > 0,$$

and term $B$ can be bounded as follows

$$B = \prod_{i=1}^{c} (1 + \frac{b}{a-c-1+i}) = \exp(\log(\prod_{i=1}^{c}(1 + \frac{b}{a-c-1+i})))$$

$$= \exp(\sum_{i=1}^{c} \log(1 + \frac{b}{a-c-1+i})) \leq \exp(\sum_{i=1}^{c} \frac{b}{a-c-1+i})$$

$$\leq \exp(\sum_{i=1}^{a-1} \frac{b}{i}) \leq \exp(b(\ln(a) + 1)).$$

Then, combining term $A$ and $B$ together:

$$K^{-2} \cdot K^{-c+2} \cdot \exp(b\ln(a) + 1) \cdot c \leq K^{-2}$$

$$\Leftrightarrow \quad \exp(\ln(K^{c-2}/c)) \geq \exp(b\ln(a) + 1)$$

$$\overset{(a)}{\Leftarrow} \quad \exp(\ln(K^{c-2+\alpha})) \geq \exp(b\ln(a) + 1)$$

$$\Leftrightarrow \quad c - 2 + \alpha \geq \frac{b\ln(a) + 1}{\ln(K)}$$

$$\Leftrightarrow \quad c \geq \frac{b\ln(a) + 1}{\ln(K)} + 2 - \alpha,$$

where the development $(a)$ is due to $c = \max\{K^{-\alpha}, \frac{b\ln(a)+1}{\ln(K)} + 2 - \alpha\}$ for $\alpha \geq 0$.

As a result, we have

$$p_k \cdot p_{\text{Beta}}(k/K) = K^{-2} \cdot p_{\text{Beta}}(k/K; a - c; b)$$

$$\Rightarrow \sum_{k=1}^{K} p_k \cdot p_{\text{Beta}}(k/K; a - c; b) \leq \sum_{k=1}^{K} p_{\text{Beta}}(k/K; a - c; b)/K^2 \leq 1.$$

□

(1)
$$\sum_{k=1}^{K} k^{-c} \geq \int_{1}^{K+1} k^{-c} dk = \left. \frac{k^{-(c-1)}}{-(c-1)} \right|_{k=1}^{K+1} = \frac{(K+1)^{-(c-1)}}{-(c-1)} - \frac{1^{-(c-1)}}{-(c-1)} = \frac{(K+1)^{-(c-1)} - 1}{-(c-1)},$$

where the inequality is due to the left Riemann sum for the monotonically decreasing function $k^{-c}$ with $c > 0$.

