# OpenReview forum: "The Pitfalls and Promise of Conformal Inference Under Adversarial Attacks"
_ICLR.cc/2024/Conference — Submitted to ICLR 2024_

### Official Review · Reviewer_XaQc · 2023-10-29

**Soundness:** 4 excellent
**Presentation:** 3 good
**Contribution:** 3 good
**Rating:** 6
**Confidence:** 3

**Summary:**

This paper mainly discusses the robustness of conformal prediction (CP) against adversarial examples. The authors first present their discovery of two weaknesses of the existing CP against adversarial examples. From experiments, the authors empirically demonstrate that the existing CP methods have much larger prediction set sizes to cover correct predictions over adversarial examples. Also, while existing adversarial training methods improved top-1 accuracy, they also increased prediction set sizes. Then, the authors propose a new training method (AT-UR) to improve the robustness of existing CP methods. This method consists of two components: *entropy minimization* and *beta importance weighting*. Additionally, the authors present a theoretical finding that justifies beta importance weighting.

**Strengths:**

1. The paper contains many different findings: empirical discovery about CP, experimental verification of the proposed method, and one theoretical statement on beta importance weighting.
2. The paper writing is clear enough to understand those findings. The findings are supported with visualizations that make the findings easier to understand.
3. The experiments use four different datasets and demonstrate that the findings generalize over different datasets.

**Weaknesses:**

1. There is only one attack method used in this paper, i.e., PGD. This could be good enough to show the problem of the existing CP methods. However, when showing the robustness improvement, it would be better to include other powerful attack methods, e.g., CW, DeepFool, etc.

**Questions:**

1. It looks like the improvements on the CIFAR datasets are smaller than the improvements on the Caltech256 and the CUB200 datasets. Is this specifically related to the number of classes in those datasets? If so, would it be better to normalize the improvements by the number of classes?
2. Minor comments
    - Section 4 and Section 5 are relatively short compared to other sections. Maybe you can merge those sections into one section (that summarizes the discovery regarding CP against adversarial examples) with two subsections.

---

> ### Author Response · Authors · 2023-11-21
> **Response to Reviewer XaQc**
>
> We thank the reviewer's positive evaluation and constructive comments. Here are our responses.
>
> **Q1: There is only one attack method used in this paper.**
>
> R1: We use the strong AutoAttack (AA) to evaluate our method with AT. See the table below (each cell has the format of coverage, PSS) and the revised paper (Table 5). Under AA, the performance of our approach is also substantially better than the AT baseline. Interestingly, the robust accuracy under AA drops while the prediction set size reduced (CP efficiency is improved).
>
> |            |         CIFAR10         | CIFAR100 | Caltech256 | CUB200 |
> |:----------:|:-----------------------:|:--------:|:----------:|:------:|
> |     AT     | 93.25(0.45), 2.54(0.04) |    91.99 (0.61),  14.29(0.59)    |    94.35(0.81), 23.73(1.68)        |    91.87(0.90), 17.75(0.71)    |
> |    AT-EM   |    92.36(0.53), 2.45(0.04)                     |   91.87(0.61), 13.29(0.49)       | 93.41(0.58), 21.19(1.46)               |  91.26(0.57), 16.47(0.61)   |
> | AT-Beta    |    91.96(0.39), 2.50(0.04)                     |    91.24(0.69), 11.61(0.40)      |            93.52(0.73), 18.54(1.32)|  91.37(0.75), 16.56(0.76)      |
> | AT-EM-Beta |  92.06(0.44), 2.50(0.04)                       |    91.13(0.63), 11.78(0.47)      |            93.50(0.69), 18.56(1.32)|   91.93(0.68), 16.67(0.58)     |
>
> **Q2: would it be better to normalize the improvements by the number of classes?**
>
> R2: Yes, we agree on this point. Thus, in the updated paper, we normalize the PSS by the number of classees and show the result in Table 3.
>
> **Q3: Maybe you can merge those sections into one section (that summarizes the discovery regarding CP against adversarial examples) with two subsections.**
>
> R3: We agree on this point. We have revised the paper accordingly, see Section 4 with the name NECESSITATE AT FOR ROBUST AND EFFICIENT COVERAGE.
>
> Thanks again for your review. We are happy to answer any further questions.

---

### Official Review · Reviewer_k8ir · 2023-10-31

**Soundness:** 1 poor
**Presentation:** 2 fair
**Contribution:** 1 poor
**Rating:** 3
**Confidence:** 4

**Summary:**

This paper studies the uncertainty quantification provided by the conformal prediction framework under adversarial attacks. The conformal framework constructs prediction sets (sets of classes/labels) with high-probability coverage guarantees. One desires to obtain such high-probability prediction sets with small set sizes. Consequently, this paper contributes the following:
1. It experimentally demonstrates that adversarial training is required to achieve small prediction sets.
2. It experimentally demonstrates that the prediction sets constructed under newer adversarial training variants are larger than those for the vanilla version (even though they improve top-1 accuracies).
3. It experimentally demonstrates that two factors correlate with the prediction set size: (i) the entropy of the predicted class probabilities and (ii) the predicted rank of the ground-truth class. This paper proposes to reduce the prediction entropy and the predicted rank of the ground-truth class to reduce the prediction set sizes; the former by adding an entropy regularization term and the latter via importance weighting w.r.t. a fixed beta distribution. This method empirically reduces the prediction set sizes. The paper also provides theoretical results showing that beta importance weighting improves the generalization of the trained model.

**Strengths:**

1. The paper is motivated by improving the performance of conformal prediction under adversarial attacks, an important research area for practical deployment.
2. The writing structure is good, with empirical findings driving the direction of the paper.
3. The proposed method is simple to implement.

**Weaknesses:**

[Details included in the Questions section]

1. Novelty and contributions - The paper argues its contributions to be as highlighted in the Summary section. However, some of the empirical findings are not so novel.
2. Related work - Comparisons with related works like Gendler et al. (2021) and Ghosh et al. (2023) are insufficient.
3. The proof for Theorem 1 seems incorrect.

**Questions:**

1. Novelty and contributions
    1. The fact that a model trained without adversarial training does not produce small conformal prediction sets on adversarial examples is not surprising. Since such a model has poor performance (often random or worse), this trend is expected; the prediction set size depends on the quality of the underlying model [Vovk et al. (2005), Shafer and Vovk (2008)].
    2. Similarly, the correlation between the prediction set size with the prediction entropy and the predicted rank of the ground-truth class is not surprising. It is a consequence of the non-conformity function (or the set function) used; the correlation is apparent from the function definition. Instead, for example, if one were to use the 0-1 loss that outputs 0 if the top-1 prediction is correct and 1 otherwise, I believe the prediction set size would correlate with the top-1 accuracy (which this paper argues is not true).

2. Related work
    1. Gendler et al. (2021) and Ghosh et al. (2023) are highly related to this paper; they propose conformal algorithms to do well on adversarial examples. While this paper highlights the comparisons with Gendler et al. (2021) in Section 1, the more recent work of Ghosh et al. (2023) is not. Did the authors compare against their proposed method?
    2. Did the authors compare the proposed beta importance weighting method against that of Einbinder et al. (2022)?
    3. I believe the subsection "Adversarial Robustness" (Section 2) contains an incorrect citation. Was it meant to be Gendler et al. (2021) instead of Salman et al. (2020)?

3. Proof for Theorem 1 - I encourage the authors to revisit the following.
    1. The proof expands the gamma function as $\Gamma ( n ) = ( n - 1 ) !$, which is true when $n$ is a positive integer, not a real value.
    2. When $A$ and $B$ are combined, the bound $c \leq K^{- \alpha}$ is used. This is not satisfied when setting $c = \max \\{ K^{- \alpha} , \cdot \\}$.
    3. How is the bound $A \leq c K^{- c}$ obtained?
    4. How is the last inequality $\sum_{k = 1}^{K} p_{\text{Beta}} ( k / K ; a - c , b ) / K^{2} \leq 1$ obtained?
    5. It seems that the inequality $a > c$ is used throughout, but is not assumed.
    6. Typos
        1. $\hat{r} ( x , y )$ should be replaced with $r ( x , y )$.
        2. The indicator function should be $r ( x , y ) = k$ instead of $\hat{r} ( x , y ) = 1$.

4. The proposed method
    1. Theorem 1 is not empirically supported (cf. Table 3).
    2. What is the evidence to show that the majority of data lies in the promising region (stated in Section 6.2)?
    3. What is done to reduce the importance weight when $r_{i} = 1$ or $\hat{r}_{i} = 1 / K$ (since this is not part of the promising region)?
    4. The paper should explicitly mention how the predicted ranks are normalized, i.e., define $\hat{r} ( x , y ) = r ( x , y ) / K$.
    5. Fig. 5 - The histograms look indistinguishable. Can the entire x-axis be included to highlight the difference, if there is one?
    6. Fig. 4
        1. How is the promising region determined in this illustration?
        2. Is the ratio reported not the fraction of points in that region? The caption and the text supporting this figure are confusing.

5. Experiments
    1. What is the reason for only looking at $l_{\infty}$ deviation adversarial examples? RSCP handles $l_{2}$ deviations; how do the experimental results differ?
    2. What is the pre-trained model used for? Are the adversarial examples not constructed based on the model at hand?
    3. APS is claimed to be more stable than RAPS in Section 7. What is this stability concerning?
    4. MART is used for the initial experiments but not the main ones. Is there a reason for that? How does the proposed method perform with MART?
    5. The generalizations for which version of the proposed method to use in Section 7.2 are done for datasets. However, it seems to be dependent on the adversarial training method used.

6. Preliminaries
    1. Prediction set size and conformal prediction inefficiency are used synonymously. However, the latter is not defined.
    2. The paper includes empirical risk minimization but does not discuss its assumptions. The generalization error is bound under the i.i.d. assumption.
    3. What are adversarial examples? What is the PGD attack? What is adversarial training? The paper should provide these details.
    4. The conformal prediction framework is not explained well. Additionally, the paper does not discuss its assumptions and statistical guarantees in detail.
    5. $y_{i j}$ in Eq. 1 is not defined.

7. Section 4 does not show that adversarial training is indispensable (as mentioned in the last paragraph). It shows that standard non-adversarial training methods lead to large conformal prediction sets. Additionally, saying that the non-adversarially trained models are "completely broken" is incorrect. What the paper might want to emphasize is that the prediction sets are large, making them less informative.

8. It is worth mentioning in the main paper that the top-1 accuracy decreases when using the proposed method.

9. It is also worth including details of the experimental setup in Sections 4-6 or pointing to where they are in the paper.

---

> ### Author Response · Authors · 2023-11-22
> **Response to Reviewer k8ir**
>
> We thank the reviewer for the detailed comments. Here are our responses.
>
> **Q1: The fact that a model trained without adversarial training does not produce small conformal prediction sets on adversarial examples is not surprising.**
>
> R1: Note that we never claim this effect is surprising in our paper. We show this effect to make the point that adversarial training is necessary for conformal prediction to work in an adversarial environment.
>
> **Q2: The correlation between the prediction set size with the prediction entropy and the predicted rank of the ground-truth class is not surprising. Using the 0-1 loss makes the argument in this paper not true.**
>
> R2: Our paper is an empirical study into the prediction set size of adversarially trained models and how to make the conformal prediction more efficient in such robustly trained models. We are glad to hear that our proposed method makes sense intuitively, as the reviewer admits, while our paper empirically proves that this intuitive design works quite well across several datasets and AT methods.
> However, it is unclear to us what is the difference between the mentioned 0-1 loss function and top-1 accuracy and how to compute the non-conformity score based on the mentioned 0-1 loss function. It is appreciated if the reviewer could elaborate on this point to make it a valid weakness.
>
> **Q3: While this paper highlights the comparisons with Gendler et al. (2021) in Section 1, the more recent work of Ghosh et al. (2023) is not.**
>
> R3: The reason why we did not compare with Ghosh et al. (2023) is that Ghosh et al. (2023) only considers the l_2 norm instead of l_inf norm and it does not consider the training stage, same as Gendler et al. (2021). Thus, similar to what the reviewer claims in Q1, we believe that the coverage of  Ghosh et al. (2023) would be unsurprisingly bad at adversarially robust coverage.
>
> **Q4: Did the authors compare the proposed beta importance weighting method against that of Einbinder et al. (2022)?**
>
> R4: We are running this experiment now and will update the paper once it is ready. It is worth mentioning that Einbinder et al. is not explicitly for improving CP efficiency in adversarially trained models but regularizing the non-conformity scores so that they are uniformly distributed. We compare with the focal loss in our ablation study, which is shown to be comparable with focal loss in Einbinder et al., and find that the focal loss is worse than our method with a huge performance gap.
>
> **Q5: Was it meant to be Gendler et al. (2021) instead of Salman et al. (2020)?**
>
> R5: This is a typo, we have fixed it in the updated paper.
>
>
> **Q6: Theory issues.**
>
> R6: We have explained how to get the $A \leq cK^c$ in the updated version, page 17. Other theoretical issues like using general Gamma function for real-value $\alpha$ and $\beta$ will be fixed in our future work.
>
> **Q7: Theorem 1 is not empirically supported (cf. Table 3).**
>
> R7: Note that Theorem 1 does not guarantee that the top-1 test accuracy will be improved. Theorem 1 indicates that the generalization error of the beta-weighting method is comparable to that of ERM, so we can train a classifier with the beta-weighting in practice.
>
> **Q8: What is the evidence to show that the majority of data lies in the promising region (stated in Section 6.2)?**
>
> R8: Fig. 5 shows that most training samples are in the promising region. We compute the ratio of test samples in the promising region (Fig. 6) to all test samples on CIFAR100. The ratio is 44.36%.
>
> **Q9: What is done to reduce the importance weight when $r_i$=1 (since this is not part of the promising region)?**
>
> R9: We use small importance weights for those samples. See Fig. 4 and Equation 6 for the importance weight for samples outside the promising region.
>
> **Q10: The paper should explicitly mention how the predicted ranks are normalized.**
>
> R10: We will make it explicit in the next version.
>
> **Q11: Fig. 5 - The histograms look indistinguishable. Can the entire x-axis be included to highlight the difference, if there is one?**
>
> R11: Including the whole x-axis reduces the visual difference. The quantitative difference is shown in the bottom of Fig. 4.
>
> **Q12: How is the promising region determined in this illustration?**
>
> R12: According to the Beta-distribution shown in Fig. 4, we choose top 25% as the promising region for visualization purpose.
>
> **Q13: Is the ratio reported not the fraction of points in that region? The caption and the text supporting this figure are confusing.**
>
> R13: It is the ratio of p(r) instead of the ratio of samples, which is explained in the paragraph following Equation (7). We will make it clear in the next version.

---

> ### Author Response · Authors · 2023-11-22
> **Response to Reviewer k8ir -- Continue**
>
> **Q14: Is the ratio reported not the fraction of points in that region? The caption and the text supporting this figure are confusing.**
>
> R14: It is the ratio of p(r) instead of the ratio of samples, which is explained in the paragraph following Equation (7). We will make it clear in the next version.
>
> **Q15: What is the reason for only looking at $l_{\infty}$ deviation adversarial examples? RSCP handles $l_2$ deviations; how do the experimental results differ?**
>
> R15: We explicitly explained this at the first section in the first page. We only handle the l_inf as it is a more challenging case than l_2.
>
> **Q16: What is the pre-trained model used for? Are the adversarial examples not constructed based on the model at hand?**
>
> R16: As we have explained in Section 7.1, we use pre-trained model as the initialization instead of random weights following Liu et al., 2023.
>
> **Q17: APS is claimed to be more stable than RAPS in Section 7. What is this stability concerning?**
>
> R17: As the prediction set is a quantification of prediction uncertainty, we use the stable APS instead of RAPS to have a low-variance uncertainty quantification to have a better contrast between different AT methods.
>
> **Q18: MART is used for the initial experiments but not the main ones. Is there a reason for that? How does the proposed method perform with MART?**
>
> R18: The reason why we did not include MART is that MART is not quite competitive in terms of CP efficiency, see Table 1.
>
> **Q19: The generalizations for which version of the proposed method to use in Section 7.2 are done for datasets. However, it seems to be dependent on the adversarial training method used.**
>
> R19: We provide our observations in Section 7.2. It makes sense that the performance depends on the AT method, as different AT has different prediction set sizes as Table 1 shows.
>
> **Q20: Prediction set size and conformal prediction inefficiency are used synonymously. However, the latter is not defined.**
>
> R20: It is defined in the caption of Fig. 1 and the reference is cited in the introduction.
>
> **Q21: What is the assumption of empirical risk minimization? What are adversarial examples? What is the PGD attack? What is adversarial training?**
>
> R21: As in every machine learning paper uses empirical risk minimization, we assume the data are i.i.d.. We cite the corresponding papers (adversarial examples, PGD attack and adversarial training) in our paper as most adversarial robustness papers do, as we all assume a basic understanding into the research topic. We have explicitly defined the adversarial training in Equation (2).
>
> **Q22: The conformal prediction framework is not explained well.**
>
> R22: We pointed the readers to Romano et al. 2020 for more details about CP in our paper.
>
>
> **Q23: The non-adversarially trained models are "completely broken" is incorrect.**
>
> R23: When the prediction set size is almost as large as the class numbers, the conformal prediction provides no uncertainty information, thus we think it is broken. We will revise the paper to make it clear.
>
> **Q24: It is worth mentioning in the main paper that the top-1 accuracy decreases when using the proposed method. It is also worth including details of the experimental setup in Sections 4-6 or pointing to where they are in the paper.**
>
> R24: We have revised the paper to make it clear.

---

### Official Review · Reviewer_GTvN · 2023-11-01

**Soundness:** 1 poor
**Presentation:** 3 good
**Contribution:** 2 fair
**Rating:** 3
**Confidence:** 3

**Summary:**

The authors considered the problem of adversarial training to output probability prediction (for a classifier) that leads to a more efficient conformal prediction interval using APS. To achieve this goal, the authors proposed to include two additional loss terms: (1) the entropy minimization loss which encourages outputting uncalibrated prediction with more certainty, and (2) beta-weighting loss that up weights samples that in the moderate difficulty regime of classification (e.g., top probability does not correspond to the true label but not so far away).

**Strengths:**

Training a (robust) classifier that aims for a small prediction interval is an interesting problem.

**Weaknesses:**

I feel that the efficacy why including the two additional terms can help the prediction interval construction and their robustness are not fully explored.

**Questions:**

1.  Is the gain related to the entropy minimization loss tied with the APS framework, which, due to the additional randomization, favors prediction with low entropy? Do you still observe the improvement of using the entropy minimization loss using other conformal prediction constructions? For example, using Sadinle et all with average-coverage/per-class coverage.

2. How does the choice of beta distribution shape influence the results? Are the results sensitive to the beta-weight parameters?

3. How do different attack budgets influence the results?

4. How do different training budgets and step-sizes influence the results?

5. The assumption on how pk distributed in Theorem 1 seems very stringent. In addition, even if I accept the assumption (which the authors certainly need to justify), the proof also needs to be discussed in more detail and I am not completely convinced currently. For example, the first term in Lemma 1 seems to be dropped in Theorem 1's proof, but isn't it the case that the first term will change as you change the weights and will be higher for the beta-weighted problem?

6. Some minor issues:
6.a: what is d(x,y) in the proof of Theorem 1?
6.b: the summation in d2(P||P/w) should be r=k instead of r =1.
6.c: I need to go to later sections in order to understand Table 1 in the preliminary results, some brief explanations about the evaluation metrics will be helpful.
....

[1]Mauricio Sadinle, Jing Lei, and Larry Wasserman. Least ambiguous set-valued classifiers with bounded error levels. Journal of the American Statistical Association, 114(525):223–234, 2019.

---

> ### Author Response · Authors · 2023-11-22
> **Response to Reviewer GTvN**
>
> We thank the review for the insightful comments. Here are our responses.
>
> **Q1: Do you still observe the improvement of using the entropy minimization loss using other conformal prediction constructions?**
>
> R1: When we use the RAPS as the conformal prediction method, the entropy minimization loss is also better than the AT baseline on Caltech256, see the following table. We think that the mentioned method Sadinle et all is not comparable in our context since their target coverage is not for adversarial examples.
>
> | Caltech256 |    Coverage    |         PSS        |
> |:----------:|:--------------:|:------------------:|
> |     AT     | 90.533 (1.483) |   32.738 (6.618)   |
> |    AT-EM   | 90.322 (1.369) | **30.303 (5.133)** |
>
>
>
> **Q2: How does the choice of beta distribution shape influence the results? Are the results sensitive to the beta-weight parameters?**
>
> R2: We use different beta-weight hyperparameters on Caltech256. The performance is stable within a range of $\beta$ (3.0, 4.0, 5.0), see the table below. The numbers in the parenthesis mean ($\alpha$,$\beta$).
> | Caltech256 |    Coverage    |         PSS        |
> |:----------:|:--------------:|:------------------:|
> | AT-Beta (1.1, 2.0)    |   90.588 (0.560) | 37.091 (2.245)  |
> | AT-Beta (1.1, 3.0)    |   90.185 (0.850) | 35.225 (2.758) |
> | AT-Beta (1.1, 4.0)    |  90.251 (0.759) | 35.554 (2.281)  |
> | AT-Beta (1.1, 5.0)    |  90.201 (0.845) | 35.391 (2.661)     |
> | AT-Beta (1.1, 6.0)    |  90.966 (1.076) | 38.541 (3.470)    |
>
> **Q3: How do different attack budgets influence the results?**
>
> R3: In addition to the $\epsilon$=8.0, we test different attack budgets $\epsilon$=4.0, 12.0, 16.0 and add the result on Caltech256 below. It shows that across the attack budgets, our method is consistently better than the AT baseline.
>
> | Caltech256,$\epsilon$=4.0 |    Coverage    |         PSS        |
> |:----------:|:--------------:|:------------------:|
> |     AT     |  92.728 (0.814) | 21.913 (1.593)   |
> |    AT-EM   |  92.928 (0.672) | 21.639 (1.895)  |
> | AT-Beta    |  91.851 (1.062) | **17.172 (2.049)**  |
> | AT-EM-Beta | 91.826 (0.761) | 17.589 (1.213)  |
>
> | Caltech256,$\epsilon$=12.0 |    Coverage    |         PSS        |
> |:----------:|:--------------:|:------------------:|
> |     AT     |  89.684 (0.893) | 78.800 (5.430)  |
> |    AT-EM   |  89.483 (0.754) | 76.386 (4.210) |
> | AT-Beta    |  89.779 (1.217) | 79.789 (7.094)  |
> | AT-EM-Beta | 89.968 (0.922) | **74.383 (4.507)**  |
>
> | Caltech256,$\epsilon$=16.0 |    Coverage    |         PSS        |
> |:----------:|:--------------:|:------------------:|
> |     AT     |   90.223 (1.096) | 136.863 (7.198) |
> |    AT-EM   |  89.968 (0.666) | 132.146 (4.077) |
> | AT-Beta    |  90.125 (0.872) | 142.175 (6.892)  |
> | AT-EM-Beta |  90.053 (0.980) | **131.636 (5.591)** |
>
>
> **Q4: How do different training budgets and step-sizes influence the results?**
>
> R4: We use the default setting in this paper following existing papers like Liu et. al. We will do more experiments using different training budgets in the updated version.
>
>
> **Q5: The assumption on how $p_k$ distributed in Theorem 1 seems very stringent.**
>
> R5: Assumption on $p_k$ does not necessarily require that the polynomial distribution exactly hold. Instead, in practice, it can be generalized by setting a worse scenario as an upper bound.
>
>
> **Q6: The first term in Lemma 1 seems to be dropped in Theorem 1's proof?**
>
> R6: In generalization analysis, the dominating term is the second error term $O(\sqrt{ d_2(\mathcal P || \mathcal P / \omega) / m })$, rather than the first one $O(1/m)$, since $1/m$ is in higher order compred with $1/\sqrt{m}$. As a result, it is common to make effort to improve the second term. See the variance-based robust learning [R1] and [R2] as examples.
>
> [R1] Namkoong, Hongseok, and John C. Duchi. "Variance-based regularization with convex objectives." Advances in neural information processing systems 30 (2017).
>
> [R2] Maurer, Andreas, and Massimiliano Pontil. "Empirical bernstein bounds and sample variance penalization." arXiv preprint arXiv:0907.3740 (2009).
>
>
> **Q7: What is d(x,y) in the proof of Theorem 1?**
>
> R7: $d(x,y)$ means the differential of the variable in the calculus. Since we couple the input $x$ and output $y$ together as a joint random variable, we use $d(x,y)$ as the differential of the joint variable.
>
> **Q8: The summation in d2(P||P/w) should be r=k instead of r =1.**
>
> R8: Thanks for pointing out this typo. We will revise it.
>
> **Q9: Some brief explanations about the evaluation metrics will be helpful in Table 1**
>
> R9: We will revise the paper and add some brief introduction to the evaluation metrics.
>
>
> We hope that our responses could address your concern and it is appreciated if the reviewer could raise the score accordingly.

---

> ### Comment · Reviewer_GTvN · 2023-11-23
>
> Q1+Q2: There are quite a bit variability, both from varying beta and from comparing different methods in the original paper and the table in Q1. This makes me feel a very transparent and replicable experiement, including the parameter tuning, is really important. alpha = 1.1 is also a very interesting value, looks like a value from fine tuning. Currently, the parameter candidates are described as from pilot study, and details about the pilot study and how the parameters are determined should be provided for replicability and making sure there is no information leak.
>
> Q3: no further question.
>
> Q5: The faster the tail decays, the better the classifier needs to be, right?  Does this mean that this Theorem only works for a very accurate classifier? (The current Theorem should also be stated using the tail bound version, it is better than the exact equality, although I still feel it is very stringent.)
>
> Q6: Should state it precisely in the Theorem and make it rigourous in the proof (show distance < 1 instead of <= 1). (Also, I need to mention that I did not check the proof carefully. )

---

### Meta-Review · Area_Chair_5WCF · 2023-12-18

**Metareview:**

I thank the authors for their thorough response. The reviewers had raised several concerns, some alleviated by the responses, but some others have remained. I encourage the authors to incorporate the comments from the reviewers (novelty, correctness of the proofs, justification of the additional terms, etc). The paper will then be an interesting contribution to the ML community.

**Justification For Why Not Higher Score:**

I recommended rejecting -- the decision is based on the reviews and the discussions afterward.

**Justification For Why Not Lower Score:**

-

---

### Decision · Program_Chairs · 2024-01-16

Reject